# Provision of Assistive Technology for Students with Disabilities in South African Higher Education

**DOI:** 10.3390/ijerph18083892

**Published:** 2021-04-08

**Authors:** Sibonokuhle Ndlovu

**Affiliations:** Ali Mazrui Centre for Higher Education Studies, University of Johannesburg, Johannesburg PO Box 524, South Africa; sndlovu@uj.ac.za

**Keywords:** assistive technology, assistive devices, students with disabilities, intersectionality, context, South African higher education, disability staff members, learning, enable and constrain

## Abstract

This paper used the Critical Disability Theory (CDT) to analyse the provision of assistive technology (AT) and assistive devices at an institution of higher education in South African. In this empirical study, data were collected through interviews with students with disabilities and Disability Rights Centre staff members. The paper sought to explore the effectiveness of the provision of AT and assistive devices, in terms of enabling students with disabilities’ learning. The provision was deemed inadequate, and a specific AT and assistive device was inaccessible to one category of disability, consequently limiting learning. The paper concludes that the provision of assistive devices at the institution enabled students with disabilities’ learning, however, there was a need for improvement by way of the Universal Design for Learning (UDL). The UDL will help all diverse students, including students with disabilities in all their categories of disability, to be assisted to learn through the provision of AT and assistive devices. It is hoped that the paper will contribute to contemporary debates on the provision of AT and assistive devices for people with disabilities in low-resource settings, from a South African context specifically, and in higher education broadly.

## 1. Introduction

The inclusion of all students in education broadly, and in learning in particular, has become a concern in higher education globally. Countries have ratified the international legal instruments, such as the Universal Declaration of Human Rights [1] and the Convention on the Rights of Persons with Disabilities (UNRPD) [2], to commit to inclusive education for all. From the Convention on the Right of Persons with Disabilities, assistive technology (AT) has also been recognised as a human right [1]. It suggests that by ratifying the UN Convention, countries automatically endorse the use of AT to facilitate all persons’ education, including those with disabilities. In essence, governments internationally, in Africa, and South Africa specifically, have committed to facilitating education by enabling everyone to gain access to learning, including persons with disabilities. The responsible stakeholders in South African higher education are also making every effort to include students with disabilities in education and learning through AT.

By way of definition, AT refers to products with the primary purpose to sustain individuals’ functioning and independence to promote their academic, social and physical wellbeing [3]. Assistive devices are also part of AT and include iPods, iPads, computers, and PowerPoint, among other [3]. One would argue that depending on the category and severity of disabilities, AT and assistive devices can enable and empower peoples’ functionality. Persons with disabilities typically confront challenges of functionality resulting from their impairments, as well as physical and social environments that are inaccessible, thereby restricting their functionality. In this paper, the terms AT and assistive devices are used together because they intrinsically relate, and cannot be separated as different entities.

The provision of AT and devices could be one way to support and enable students with disabilities’ learning in higher education. All students who gain access to higher education would have met particular programmes’ stipulated entry requirements, reflecting their intellectual capability. However, depending on the category and severity of disability, limitations in learning are imposed by both environments and individual impairments. By providing students with disabilities appropriate AT and devices, they could access learning as easily as those students without disabilities [4,5,6,7]. A systemic review of the use of AT and devices by students with disabilities in higher education has revealed that it has significant positive impacts on their academic engagement, psychological wellbeing and social participation [3]. The argument presented is that AT and devices increase the performance of academic tasks and enhance learning, improve functionality, reduce activity limitations, promote social inclusion, and increase participation in education and the labour market [3]. From the assertion, it appears that AT and devices add significant value to the general performance of students with disabilities in higher education, and enable them to do what they might not have done otherwise. It therefore appears that AT and devices could indeed be useful in enabling students with disabilities’ access to learning in higher education.

### 1.1. Provision of AT and Assistive Devices in the United Kingdom

The provision of AT and devices in the United Kingdom (UK) specifically is important in this paper because its higher education system is used as a benchmark for South Africa in terms of the inclusion of students with disabilities [8]. In the UK, after assessment, eligible students with disabilities are provided with AT and devices through the Disabled Students Allowance (DSA) fund [9]. The DSA is the fund provided by the government through various funding authorities at institutions of higher education. It is used in purchasing mainly AT and assistive devices, such as screen readers or text-to-speech systems, speech recognition software, alternative input devices, and is used in paying human resources such as note-takers and interpreters. The DSA also provides an allowance to cover access to the internet, Braille papers, photocopying costs, and additional travelling costs incurred by students with disabilities [10]. It could be argued that higher education and the UK’s provision of AT and devices is effective because there is appropriate implementation of policies [11], which hold the responsible authorities accountable in their task.

### 1.2. Provision of AT and Assistive Devices in South African Higher Education

Like the UK, South African higher education also provides AT and assistive devices to enable persons with disabilities’ to access education. However, higher education institutions are not yet fully transformed to include all diverse students in learning [12], particularly those with disabilities. On introducing inclusive education in democratic South Africa, a minister stated in the White Paper: “Let us work together to nurture our people with disabilities so that they also experience the full excitement and the joy of learning and to provide them, and our nation, with a solid foundation for lifelong learning and development” [13] (p. 4). One would argue that by way of fulfilling the minister’ claims, the South African context of higher education makes every effort to consider students with disabilities. The provision of AT and devices is one of the ways through which persons with disabilities are assisted to function optimally. AT and devices are provided through disability units and are sourced through the South African DSA. In South African higher education, the DSA is a fund that is part of the National Student Financial Aid Scheme (NSFAS), and while the NSFAS is a loan for disadvantaged students, for students with disabilities it is a grant [14].

In southern Africa largely and South Africa specifically, AT and devices mainly focus on mobility, hearing, vision, communication and cognition limitations [15]. Examples of AT and assistive devices include, among others, prosthetics, hearing aids, spectacles, white canes, and wheelchairs [16]. Thus, various low and high-tech AT and devices are provided in higher education institutions in South Africa. However, this paper discusses the provision of AT and assistive devices in terms of enabling access to classroom learning, in which some of the listed devices do play a role.

It has been determined that although higher education makes provision for AT and assistive devices, students with disabilities continue to be excluded from learning. A number of studies have been conducted to understand the continued exclusion of students with disabilities in teaching and learning in South African higher education [17,18,19,20]. This particular study comes from a unique angle of using specific theoretical concepts drawn from Critical Disability Theory to understand the provision of AT and devices, as a way to promoting and enhancing students with disabilities’ learning.

### 1.3. South African Higher Education as A Low-Resource Setting

South Africa has the most comprehensive policies of inclusion among all African countries [21]. The country has a strong political will for inclusion, as seen in its Constitution [22], legislations of non-discrimination, including the Employment Equity Act, No 55 of 1998 [23], the Promotion of Equality and Prevention of Unfair Discrimination Act, No 4 of 2000 [24], and policies of inclusion (White Paper 6: Special Needs Education: Building an Inclusive Education and Training System [25], and the Education White Paper 3: Transformation of the Higher Education System [26]). However, in this paper, South Africa’s higher education system is considered a low resourced setting since the implementation of its policies is slow and poorly executed [27]. The relevant policies have managed to increase historically disadvantaged social groups’ formal access into higher education after the demise of apartheid [28], yet students from privileged and disadvantaged backgrounds continue to experience a lack of access and success in higher education. The reason is traced to the articulation gap between schooling and higher education largely [29]. The lack of access and success are exacerbated for students with disabilities because for them, every challenge experienced by other students, is doubled. Students with disabilities experience the same obstacles as all other students, with the addition of impairment-related barriers.

In the South African context of higher education, barriers for students with disabilities come in different shapes and sizes. Among others, are the inaccessible infrastructure [30], barriers of high entry-level requirements into specific professional programmes [13], exclusive classroom teaching practices [31,32], an unwillingness among academic staff to include these students [33], and a lack of relevant and adequate AT and assistive devices, resulting in a digital divide [34]. Some institutions cannot even afford to accommodate some disabilities such as hearing impairment because of a lack of technological resources, due to inadequate funding [35]. Thus, inadequacies in terms of access and success for all students, including those with disabilities, is topical in the current debates on South African higher education, qualifying the country as a low resourced setting.

Though efforts are made to provide AT and assistive devices to students with disabilities to enable them to learn, the way in which provision is made defeats the intended purpose. Students with disabilities thus continue to confront limitations in their learning despite being provided with some AT and devices. The main research question of the study was: Is the way in which AT and assistive devices are provided, enabling students with disabilities’ access to learning in the programme of Education at the institution? Secondary research questions were also asked to assist in answering the main question: (a) Who decides what AT and assistive devices should be provided at the institution? (b) Do the AT and assistive devices that are provided enable students with disabilities’ learning?

## 2. Critical Disability Theory as Framework

Critical Disability Theory (CDT) is the theoretical framework that was employed to underpin this paper. This theory was a development from the Meta-Critical Theory [36] and its proponents comprise a range of post-conventionalists, post-structuralists, and post-colonialists who draw most of their ideas from the foundational work of Michel Foucault, Judith Butler and Jacques Derrida [37]. They built their ideas on conventional disability studies, and the way in which disability has previously been conceptualised. The proponents acknowledge the pioneers in disability work’s achievements, but constructively critique and problematise specific disability issues. Their interest is in generating new ways of understanding disability as one of the identity markers, just like race and gender [38]. They argue that persons with disabilities are undervalued and discriminated against [37] and their aim is to improve the livelihoods of persons with disabilities specifically, and all diverse persons generally. Thus, one would argue that the CDT is about the inclusion of all diversity in mainstream society, including those with disabilities.

The theory explains issues of disability to empower and emancipate those with disabilities. It falls under the umbrella that incorporates theories such queer theory, critical race theory, and/or feminism. Like the other theories, it seeks to critique established, hegemonic and narrow understandings, in this case with regard to disability and the oppression of persons with disabilities. It is a bottom-up theory [39], which involves creating opportunities for persons with disabilities to share their lived experiences of disability and ensure their voices are heard. It posits that disability is a form of diversity, and this extends beyond an outmoded understanding of disability as an identity for persons with impairments [40]. Critical disability theorists have deconstructed the dualist dichotomy of ‘disabled and non-disabled’, arguing that such a construct casts one group as superior and the other as inferior [41]. This inevitably results in one group dominating the other. The CDT can thus facilitate the emancipation of the oppressed [42,43] when it is employed to further the objective of the transformative paradigm.

The CDT is focussed on understanding issues of disability in the Global South perspective and non-Western settings [44]. Proponents of the CDT seek to shift the understanding of issues of disability specifically, and inclusion and exclusion in general, from that of a Eurocentric Global West perspective, to include voices from the South [45]. The theory is thus primarily concerned with privileging the voice from the Global South, to ensure it is also heard [45]. The CDT considers the material and local contextual conditions, primarily questioning the social practices and structures that influence certain social groups’ marginalisation, including those with disabilities [37]. As this study was conducted at a South African institution of higher education, the theory’s application could reveal issues that are particular to the Global South by way of geographical location.

For this paper, intersectionality and context have been drawn from the CDT as two theoretical concepts that are relevant to underpin and explain the provision of AT and assistive devices at an institution of higher education in South Africa. This will help explain and draw an understanding of the South African higher education system, particularly the structures and practice of teaching and learning.

### 2.1. Intersectionality

Intersectionality is useful in disrupting the notion that students with disabilities are a homogenous social group with ‘special needs’. Critical disability scholars highlight intra/intersectionality to emphasise that disability intersects with other multiple identities including sexuality, race, gender, ethnicity or class. It should thus not be considered in isolation, but be placed in the centre of these identities [38]. This illustrates that students with disabilities are diverse in themselves, as influenced by their different home backgrounds, schooling backgrounds, and their unique experiences or exposure.

Intersectionality also creates an understanding that disability should be considered as fluid and ever-changing, an entity shared by all people with and without disabilities [38]. This implies that even persons without disabilities can be limited in specific contexts because disability is not a rigid category. When an individual finds themselves limited in performing at one time or another, they are ‘disabled’. Thus, students without disabilities could be as limited in their learning during a pandemic, as currently experienced with Covid-19, as those with disabilities.

Intersectionality further reflects that disability should not always be seen in terms of disadvantage. Individuals with disabilities could also be placed in positions of power [46]. Disability should be considered a springboard; a space from which to think through a host of other political and theoretical issues that apply to all identities [40]. It is further explained that the ‘disabled’ body is not only understood in the context of oppression, because persons with disabilities are intersectional subjects who also embody other positions, which can be powerful and valued in an ableist culture [46]. This is to say that some students with disabilities are not in positions of oppression, as might be overgeneralised. They have power in their other identities, or they are influenced by the power of other powerful people with whom they associate. In terms of understanding intersectionality, disability should therefore not always be viewed in light of disadvantage [47].

There has also been a shift from the view of double oppression [47] whereby disability has always been considered the intersection of one axis of oppression with another. As stated, intersectionality can privilege and does not always yield double oppression [48]. Understanding intersectionality in light of disability and privilege could disrupt the perspective of inequality based on gender and race, popularly manifesting in African societies. There could be an understanding that a White female student with a disability from a high socioeconomic class could be privileged over a Black male student with a disability from a low socioeconomic class. Thus, intersectionality can help to explain students with disabilities’ uniqueness of learning, even in cases where they have the same impairment. In this study, the theoretical concept of intersectionality could help explain how students with different categories of disabilities experience the provision of AT and assistive devices and how, as unique individuals, they are enabled or constrained in their learning by this provision.

### 2.2. Context

Context is a useful theoretical concept to inform an understanding of the provision of AT and assistive devices at an institution of higher learning. It presents an overview of how disability in itself is constructed and perceived in a specific environment [38]. Disability should be located in a context since it is conceptualised differently depending on the context within which it exists [38]. In African societies broadly, and South Africa specifically, the conception of disability is influenced by cultural tradition; as a result, persons with disabilities are often viewed as limited and associated with inability. In higher education, the negative conception results in low expectations of students with disabilities [28], even though their academic credentials facilitated their access to higher education. African society and stakeholders in South African higher education institutions have negative attitudes towards students with disabilities, resulting in the academic staff in particular being unwilling to include students with disabilities in their teaching [33]. The particular students are viewed as a burden [49]. Thus, all negativity about disability in the South African context could be linked to the context within which it is constructed and perceived. The issue of context therefore becomes critical in explaining the way in which AT and devices are provided at an institution of higher education in South Africa. By virtue of the focus on the South, the theoretical concept of context becomes important in informing an understanding of contextual differences, and the implications for the provision of AT and devices.

Besides promoting an understanding of the context within which AT and devices are provided, the concept is also important for the consideration of a specific context, when thinking about or suggesting intervention strategies for improving the provision of AT and assistive devices at the institution in particular, and higher education in general in South Africa. This is important because contexts of learning vary greatly. What works in one context may not work in another. Thus, the theoretical concept prevents over-generalisations in terms of intervention, and in this paper, it could help in locating interventions within the specific context of South African higher education, which is different from other learning contexts; socially, politically, and economically.

## 3. Methods

The researcher utilised purposive sampling [50] to select students with disabilities, who were studying a professional degree in Education, to participate in the study. Six students ultimately volunteered to participate. Disability categories for students with disabilities included visual impairment, hearing impairment, and physical disabilities. Two of the participants had visual impairment, one with total vision loss and the other with partial vision loss. Two students had both partial hearing loss, and two students with physical disabilities were in wheelchairs. The sample cut across race, gender, age and schooling background in terms of mainstream and special schools. Participants were both undergraduate and postgraduate students in the final year of their degree programme. All the student participants had disclosed their disabilities during admission and were supported by way of AT and assistive devices that were distributed by the Disability Rights Centre at the institution. This was peculiar because literature has consistently revealed that most students with disabilities do not disclose their disabilities on admission and registration in institutions of higher education because they fear stigmatisation [51,52,53]; this phenomenon is particularly common among students with invisible disabilities such as hearing impairments [54]. Thus, sampling targeted students with disabilities because only those with a lived experience of disability can attest to the effectiveness or ineffectiveness of the AT or devices to their learning.

Disability Rights Centre staff members were also sampled. The sampled included four participants with disabilities and six without. The disability categories for the four staff members were low vision and physical disabilities. Two members who had albinism had low vision and two with physical disabilities were in wheelchairs. The sample disregarded gender, race, age, work experience and position at the Disability Rights Centre. The staff members were sampled because they were the ones providing support in the way of AT and assistive devices to students with disabilities. Ten members volunteered to participate, and they were all permanent staff members of the Disability Rights Centre, which is said to be the best in the country. Thus, all 16 participants could provide rich data and promote an understand the provision of AT and devices in a higher education setting that is low resourced. To ensure participants’ anonymity and confidentiality, numbers were used as codes to identify them.

Though the study sample was small, it suited the qualitative methodology in terms of purposive sampling, which provided in-depth and relevant data on the provision of AT and assistive devices at the institution, the phenomenon that was being explored. The sample was also able to provide data that addressed the research problem and the two research questions, supporting the case-oriented analysis that was fundamental to the study [55]. It is argued that the selection of a rich information sample is fundamental in qualitative research as it promotes gathering worthwhile and relevant data to understand the phenomenon under investigation [56]. Thus, the data provided suited the scope of the study and enabled rigorous data analysis and data saturation. Also, a microscopic view was presented in terms of the problem and the research questions of the study. Sufficient results were drawn from the lived experiences of students with disabilities themselves and the staff members of the Disability Rights Centre, about the provision of AT and assistive devices and how it enabled or constrained learning.

Table 1 below show the participants’ demographic information, first of the students with disabilities and then the staff from the Disability Rights Centre.

Both student and staff participants were sampled from an institution of higher education that was formerly advantaged. This institution even accommodated students with disabilities during apartheid, and the Disability Rights Centre was established in 1985, before South Africa attained its independence. The Centre supports students with disabilities’ learning in several ways, including providing them with AT and assistive devices. The Centre is said to be the best in the country, supporting the largest number of disabilities, having the highest number of permanent staff, and was awarded a prize for being the best by the Department of Higher Education and Training in 2012 [57].

Education, as a programme of study, has a unique history in South Africa stemming from the apartheid era. Its history qualified it to be considered, by way of classification of professions, as one of the low-level professions [58]. It was previously offered in teachers’ colleges, in which situational knowledge was emphasised rather than disciplinary knowledge [59]. In democratic South Africa, there was a shift, and the programme was now offered in universities, where students had to acquire both disciplinary and situational knowledges. In this paper, the researcher’s interest was in understanding the role of AT and devices in enabling students with disabilities’ learning at the institution. However, the paper does not go deep into the learning of specific concepts but explores learning generally, at the institution and not in integrated learning settings, such as schools.

The Disability Right Centre staff members referred the researcher to students registered with them, and these students were then sampled. In accessing the students, the researcher was referred to other students with disabilities by those who first volunteered to participate. In essence, snowballing [50] was used as the main strategy to access participants.

Ethical considerations were adhered to in the study and permission was obtained from the gatekeepers at the institution, as required [43]. All participants in the study volunteered their participation. They were informed about the focus of the study, their right to withdraw from the study, not to answer any questions that made them uncomfortable, the right to end the interview at any time, and even to amend transcribed interviews where there is a misrepresentation. In addition, confidentiality and anonymity were guaranteed for those who volunteered to take part in the study [43].

Semi-structured interviews were conducted with students with disabilities and the Disability Rights Centre staff members to collect data on the provision of AT and assistive devices at the institution and how it enabled the students to access learning. This kind of interview was selected over other types because participants’ perspectives are captured in a detailed way. With this type of interview, the researcher cannot afford not to source the qualitative data they need [51]. There is also liberty to probe further when the researcher still needs more data on the phenomenon. Thus, the interviewing technique ensured that the researcher had flexibility in asking questions [50] and obtained rich data from the participants. Participants were also afforded an opportunity to offer depth in their responses [52].

Those who volunteered were asked to indicate their interest by contacting the researcher through email, and indicating a time that would be convenient for them to participate in interviews, and a place where they felt comfortable to be interviewed. The student participants were interviewed in different places they chose themselves. One student with hearing impairments was interviewed from the library lawns, which was a very quiet space. Three students were interviewed at their residences because they felt it was convenient for them as they had physical and visual limitations, respectively. One chose to be interviewed at a place outside the dining hall because she liked the area, and one was interviewed in his office, as he was also a staff member. Moreover, the staff members from the Disability Rights Centre chose to be interviewed at their offices.

The topics (questions asked) that were discussed during the qualitative interviews included: learning needs and AT and assistive devices provided at the institution; funding for AT and assistive devices; who designs AT and assistive devices; who provides the AT and assistive devices; the adequacy of AT and assistive devices at the institution; are AT and assistive devices relevant in meeting learning needs; and the effectiveness of AT and assistive devices in meeting learning needs.

Audio recording and verbatim transcription of all interviews were done. A digital recorder was used to record all interviews with the participants’ permission. Mechanical recording reduces researcher bias [60] and enables accurate verbatim transcription. All recorded interviews were then transcribed into text.

The researcher first performed data analysis independently, by reading and re-reading each individual interview many times, coding, grouping and regrouping codes until major themes emerged from the data. Sufficient data were gathered to address the research questions; this was determined through the number of major themes that emerged from the data. Patterns of recurrence emerged as data were re-read. For example, both students with disabilities and staff members mentioned challenges with the Disability Rights staff providing AT and assistive devices 16 times. The author required no fewer than 10 participants to mention the same thing 10 or more times to develop a legitimate category. Numbers were used to mark recurring patterns, and instances of the same thing being mentioned were collated and grouped together. In this way of coding and categorisation, adequate data were retrieved manually.

Three experts in qualitative research were then asked to analyse data as critical readers. The consensus of three different readers was obtained by the researcher first inferring the essence of minor themes, abstracting and collapsing them into conceptual themes. The same minor themes were sent in batches to specialists in qualitative research to also abstract them to conceptual or major themes. The three different readers’ analyses were compared and matched, and the same major themes emerged from all readers. Thus, a constant comparative approach to data was utilised, in which the researcher and other critical readers continued to compare their analyses [50]. To confirm the trustworthiness of the findings, a code-recode process and examination of the analysis were performed by peers [50], and it was confirmed that similar major themes emerged from the data.

ATLAS Ti could have been used for data analysis, but the researcher did not use any software because there were so few (only 16) interview transcripts. The researcher understood that it was her responsibility to do the analytical work, and manual analysis allowed her to gain a deeper understanding of the data as she read it repeatedly. Moreover, manual analysis allowed for the coding of data and deciding which codes link with the broader objectives, aims, and research questions. As a result, meaningful (and not automated) labelling and categorisation of data was facilitated. Repeated manual scouring of data was necessary to thoroughly interrogate the phenomenon and understand it broadly.

## 4. Results

### 4.1. Provision of AT and Assistive Devices at the Institution

The abstract themes that emerged from the data were grouped as two major themes and analysed using specific concepts from the CDT. The major themes included: (a) Institution making decisions on what AT and assistive devices should be provided to students with disabilities; (b) Provision of AT and assistive devices enabling students with disabilities’ learning.

#### 4.1.1. Institution Making Decisions on What AT and Assistive Devices Should Be Provided

According to both student and staff participants, the institution decides which AT and devices should be provided, and these are distributed through the Disability Rights Centre. There is a structure in place specifically to support students with disabilities at the institution. This structure reflects that the institution is intentional about supporting students with disabilities’ learning through the provision of AT and assistive devices.

The technician employed by the university is responsible for designing and ordering AT and assistive devices. The technician is an individual without disabilities, and the participant shared:

*If we have a new student who is coming with a disability, that is part of my job, to design technologies for new disabilities. We need to improve technology because every technology in life need a user and I as well, must always have something new in my mind* (Member 5).

From this statement, one would say that an individual is responsible for deciding what AT and assistive devices students with disabilities need for them to access learning. Interestingly, those who determine the kind of AT and assistive devices students with disabilities require, are not disabled themselves.

#### 4.1.2. Provision of AT and Assistive Devices Enabling Students with Disabilities’ Learning

It emerged from data that the AT and devices provided by the institution enable students with disabilities’ learning. Similar responses were obtained both from the Disability Right Centre staff members and students with disabilities. One member stated:

*We have devices available to students with disabilities; we do have iPads. In addition, students with disabilities have real benefitted from those devises. For example, a student will really benefit from an iPad, because they do not need to carry many books around the campus* (Member 1).

Also justifying the usefulness of AT and assistive devices for students with disabilities, a participant added:

*We cater for different disabilities and now we have someone who is using a Dragon. It is a computer that you just talk and it types, and the student will not have much of the spelling errors. The computer does the writing for the student* (Member 10).

Another participant had the same view about the AT and assistive devices provided to students with disabilities enabling their learning at the institution:

*Why people see us as number one, it is because we have been so long in the business of assisting different disabilities. Something new that we have recently got, it is a new device called eye tracker. It is for students with physical and nervous disability, who cannot use their hands to type, who cannot handle a mouse. With this device, a student uses his eyes; a student can control the mouse with their eyes. We spent about sixty thousand on it. It is something new that other universities do not have* (Member 4).

Based on responses from the Disability Rights Centre staff members, it is evident that AT and assistive devices that are provided to students with disabilities at the institution make learning easier for them; even those who have severe central nervous system conditions and cannot use their limbs.

In support, the students confirmed that those gadgets are useful in assisting their learning at the institution. In the words of one student:

*I have two impairment needs. The other impairment is scoliosis, which is the physical curvature of the spine. I am also partially deaf and because of this, the Disability Unit gave me a voice recorder to use in my lecturers. The thing was, I found it very useful because I did not have enough time to study and transcribe my lecture notes* (Student 1).

Another student agreed, and explained:

*Lecturers know how to make work accessible. Computers are taking away, helping deaf students a lot more, for example, computers are taking over someone’s head and helping him when writing notes. I think technology helps there* (Student 3).

The students themselves, who had a lived experience of disability, thus confirmed the usefulness of the AT and devices they receive at the institution, in terms of their learning. This confirmation by the students themselves cannot be contested.

While both the staff members and the students stated that AT and assistive devices were provided to those with disabilities, there was a contradiction in terms of the adequacy of the provision. Staff members stated that AT and assistive devices were adequate at the institution, while students claimed they were not. A staff member commenting on the adequacy of funding for AT and assistive devices explained:

*We are buying more adaptive devices for students with disabilities through the University fund. I am waiting for that student who says there is no fund, to tell us exactly what does he needs that he cannot get. Recently, eight got brand new laptops, some got hearing aids. We are even extending the support to a human resource, I mean an assistant who is paid from Disability Fund. DSA is more than good enough for students* (Member 4).

A student expressed a contradictory view:

*There was a time that they said we should list the assistive devices that we want, and hey, I listed, after sometime they said they did not have enough for those assistive devices. Many students had asked for assistive devices but the requests were cancelled. I think maybe funding is not enough to cater for everybody with disabilities. I really need resources and assistive devices. Like I need a Braille machine. I will be able to prepare teaching aids* (Student 4).

Another student also shared:

*The funding covers assistive devices, yes, but assistive devices are very expensive. Like when someone needs a scientific calculator, the Braille one is R4 000-00. Braille machine its R4 000-00. There was a time, I did not have a laptop, I went to the Disability Unit and they told me to go to the Financial Aid office, I went there but I did not get assistance* (Student 2).

The contradiction between staff members’ and students’ views about the adequacy of AT and assistive devices reveals that what the staff saw as adequate, students with a lived experience of disability did not see in the same way. The staff were genuine in seeing the provision of AT and assistive devices as sufficient to enable the students with disabilities’ learning. However, for those with a lived experience of the phenomenon, they are not adequate. When AT and devices are not adequate or insufficiently available, it could negatively impact learning because some students may not have access to the AT and assistive devices to help them learn. Also, the institution’s good intention of deciding what AT and assistive devices students with disabilities require to enable their learning could be defeated when somebody without a disability makes these decisions.

According to the students with disabilities and the staff members from the Centre, a variety of AT and devices are provided, including computers with JAWS, Braille machines, Kindles, magnifying glasses, ‘Dragon’ computers, eye trackers and iPads. One of the staff members, who is a technician, explained how some of the AT and assistive devices work:

*To some students I have loaned iPads because they need them. We have again have the Kindle, it is a simple device, it is almost like an iPad, a small iPad and it is handy for books* (Member, 5).

A student with disabilities who confirmed being provided with a Kindle also expressed the same experience; it enabled her to carry books, which she would not be able to carry because of the category of disability she had. She stated:

*They helped me by giving me a Kindle, which allows me to download books and then I do not have to carry heavy books around* (Student 1).

While there was a strong indication that most AT and devices assisted students with disabilities’ learning, it also emerged from the students with disabilities themselves that not all of them were assisted in their learning by all AT and assistive devices provided by the institution. Some AT and devices hindered learning in different ways. Students with disabilities shared the problems they encountered with some assistive devices, and said:

*The computers we use have screen readers. They have JAWS, but it cannot read signs, Mathematical signs and graphs. JAWS does not read pictures. And Maths also, there are signs JAWS cannot read. Sometimes they teach in PowerPoint but in my case, JAWS cannot read PowerPoint* (Student 6).

The student continued to explain how the computers limited her learning:

*As I said, learning is not accessible sometimes because they are sending notes on PowerPoint, JAWS does not read PowerPoint, and you miss those notes. In addition, this SAKAI thing, my computer does not read all things that are there, so you have to ask every time, what is happening here. It is good that we are learning through the computers but to us with visual problems, it is not so accessible* (Student 6).

Despite more students with disabilities stating that they were enabled in their learning by the provision of the AT and devices, a student with total visual loss was limited by computers with JAWS software that was not sensitive to mathematical signs and PowerPoint. Though it was just one student out of six that confronted limitations, it has a significant impact because this student was not only limited in the area of mathematics but in any learning area where PowerPoint was used.

## 5. Findings and Discussion

The study’s findings were that, firstly, the institution designs a variety of AT and assistive devices and decides which ones are relevant. They then distribute these to students through the Disability Rights Centre. Secondly, the AT and assistive devices provided at the institution promote students with disabilities’ access learning, though not all categories of disabilities were enabled because some AT and devices had their own limitations.

The researcher then attempted to understand the provision of assistive devices at the institution in the light of context and intersectionality, with the theoretical concepts drawn from the CDT. The theorisation is meant to explore the provision of AT and assistive devices to students with disabilities and the enablement or constraint of learning, not only at the specific institution, but also in other institutions with a similar context, in South Africa specifically, in Africa, and internationally. The understanding that was informed by context and intersectionality could influence interventions to improve the provision of and support related to AT and assistive devices; for students with disabilities, and in terms of creating a learning environment that is inclusive of all diverse students at the specific institution, and in higher education in South Africa generally. In this paper, the researcher further proposes the Universal Design for Learning (UDL) as an intervention that might promote learning through the provision of AT and assistive devices.

### 5.1. The Institution Making Decisions on What AT and Assistive Devices Should Be Provided

The matter of the institution deciding what provisions to make for students with disabilities in terms of AT and assistive devices could be explained according to context; the context within which disability is viewed, understood and constructed. In such a context, there is a dynamic of power, whereby the powerful decide what is right for the powerless [61]. Previous literature also reports significant instances of those without disabilities deciding, designing and implementing what they think is rightful for those with disabilities. In one study, students with disabilities themselves stated that they feel patronised by those without disabilities when they impose a particular way of enabling their access to learning on them [55].

One would argue that while AT and devices decided by the institution are enabling those with disabilities’ functioning and learning, their provision to persons with disabilities as a separate social group could be seen as informed by the individual model.

As the findings show, the AT and assistive devices provided to students with disabilities are decided by the institution, through a specific structure, the Disability Rights Centre. The staff at the Centre are responsible for the design of AT and devices and making provision to students with disabilities. Evidence shows that it is specifically the technician, who is not disabled, who decides what AT and devices to provide students with disabilities. This could be explained as a top-down approach, as opposed to the bottom-up [39], whereby the relevant AT and assistive devices are decided by those at the top, not those with a lived experience of disability at the bottom. Top-down approaches where persons without disabilities impose on those with disabilities significantly influence their exclusion in education generally [39], and in learning, particularly. As a result, the decision of what AT and devices to be provided should be left with the students with disabilities because they have a lived experience of disability [62,63]. They know exactly what is suitable for them. However, those without disabilities continue to make decisions and speak for them [64], misrepresenting and limiting their functionality and learning. In the particular mentioned case, a student with visual limitations was limited in learning mathematics through the provision of Braille machines with JAWS software that did not read mathematical signs. It stands to reason that making decisions on what AT and assistive devices should be provided, constrains learning when the AT and devices manifest certain limitations, which were only discovered when the student used them.

Besides constraint on learning, decision-making by the institution on the provision of AT and assistive devices has other broad limitations that the researcher cannot gloss over. The issue could be seen as some kind of normalisation and ableism, so that students with disabilities are also accommodated in learning. Normalisation and ableism are contested and critiqued because they manifest in the way in which society tries to make those with disabilities what they are not [58,65].

Though the provision of some AT and devices, as decided by the institution, enabled students with disabilities’ learning, one would argue that it is only an individual accommodation, and not an inclusion of all diverse students at the institution. The term ‘hospitality’ is often used in the analysis of the metaphors used in the context of individual accommodation in schooling. It is argued that when the term is used to shape and inform provisions, other social groups like those with disabilities could be seen as guests, not belonging by right [66]. The provision of AT and assistive devices is thus critiqued as a way of individual *accommodation* of students with disabilities, and not *inclusion*. Individual accommodation denotes un-belonging, whereby students with disabilities do not belong; they are there by way of the demands of the Constitution [7], non-discrimination legislations [23,24], and inclusive policies [25]. In addition, since they have formally accessed the institution, accommodation in terms of AT and devices must be provided so that they are also enabled in learning. In has also been noted that terms like ‘individual *accommodation*’ are justifiably used to make provision and offer support to students with disabilities in institutions of higher education in institutional policies, one of which is the provision of AT and assistive devices. It is argued that the way inclusion is thought about determines the way in which it is practised [67]; in this case, individual accommodation is taken for inclusion in the way AT and devices are provided.

The evidence from data reveals that the institution’s decision on what AT and assistive devices should be provided, did enable students with disabilities’ learning. Of the six student participants, only one experienced limitations in terms of JAWS software not reading mathematical signs. Two other students with visual impairments reported the software on the institution’s computers did not limit their learning but aided them. Thus, five students were enabled in their learning by the AT and devices chosen by the technician at the Disability Rights Centre. Therefore, it is evident that despite the decision on what AT and assistive devices to be provided being made by the institution, it still enabled more students with disabilities’ learning.

### 5.2. AT and Assistive Devices Not Enabling All Students’ Learning

The finding that a student with a specific category of disability was limited in her learning by AT and assistive devices that were inaccessible, could be explained in terms of the interplay of intersectionality in students with disabilities’ learning, as influenced by the provision of AT and assistive devices at the institution. Intersectionality helps in understanding that students with disabilities are not a homogeneous group [68]. Individual students with disabilities’ learning needs are different, and their impairment categories are varied, resulting in AT and devices not always meeting the needs of all six students in the Education programme. It is argued that since students are deemed intersectional, even those with the same impairments may require different support [69]. Therefore, even though the AT and devices provided enabled students with disabilities’ learning, students with a specific category of disabilities were constrained by one kind of assistive device. However, this limitation was isolated to a single student’s experience, as discussed earlier. The researcher then argues that despite intersectionality being at interplay, the provision of specific AT and assistive devices at the institution catered for more categories of disabilities and enabled their learning than those that did not.

## 6. Other issues of Relevance from Literature about the Provision of AT and Assistive Devices

Previous research has revealed that low-resource settings do not have adequate AT and assistive devices for students with disabilities. One study reported that, at the University of Zimbabwe, considered a prestigious institution of higher education in Zimbabwe, there is a lack of assistive devices for students with disabilities, and the academic staff lacked competencies and skills in using assistive devices [62]. Furthermore, students with a lived experience of disability studying at the higher education institution stated that it was difficult for them to understand the content matter taught by lecturers without assistive devices [62]. From the students with disabilities in the Zimbabwean context, it is confirmed that AT and devices assist learning, and for them, it is difficult to learn without these products.

In the South African context, previous research has also revealed a lack of adequate resources, and AT and assistive devices in Disability Rights Centres. This hindered students with disabilities’ access to learning in higher education [54]. It has also been reported that there is no adequate funding for disability units to effectively support students with disabilities’ learning [8]. With the lack of funding broadly, the adequacy of providing AT and devices may not be ascertained. In the case of the specific institution, though the Disability Rights Centre staff members stated that there is adequate AT and assistive devices, students with a lived experience of disability, disagreed. The researcher would argue that there is consistency in the lack of adequate AT and assistive devices according to previous research and the present study. Students with disabilities’ comments that they lacked some gadgets they need for their learning, are supported in literature, and the extent to which it affects their learning, could be a backdrop for future research.

The issue of AT and devices hindering students with disabilities’ learning has been raised in literature. Scholars have emphasised that the JAWS screen readers were not reading mathematical signs [70]. Therefore, students with visual impairment are limited in acquiring disciplinary knowledge in the learning area of mathematics. As practicing teachers in schools, they may counter the same limitation if screen readers with JAWS are used. It could be argued that while screen readers with JAWS seem to be an improvement on Braille, they have their own constraints on learning. In the present study, it was only one student with total visual impairment who was limited in her learning by such an AT and assistive device, yet it could have significant implications in other institutions with many students majoring in mathematics. Therefore it cannot be concluded that this has fewer implications for learning based on data available and evidence from this particular study, which is not generalisable.

Literature available in the South African context has further confirmed that the academic staff who are central in the practice of teaching and learning lack training and expertise to teach students with disabilities [71]. It is also reported that they are unwilling to attend workshops organised by disability units to learn about teaching different categories of students [54]. Moreover, it cannot be assumed that when lecturers lack training and are unwilling to teach students with disabilities, they have expertise in using AT and assistive devices to enhance learning. Academic staff’s lack of training to teach students with disabilities is not an issue in South African higher education only, but is also found in other low resourced contexts. With specific reference to the Zimbabwean higher education context, lecturers lacked training on using AT [62], reflecting that teaching staff are incompetent in teaching and using AT and devices in low resourced settings. Against this backdrop, it cannot be assumed that the mere availability of AT and assistive devices at the institution would promote access to learning for students with disabilities.

For the present study, evidence from data reflects that the provision of AT and devices to students with disabilities at the particular institution is better than other institutions in low-resource settings. Students themselves stated that their learning is enabled by the provision of AT and assistive devices. However, since the study is of limited scope and cannot be generalised to other wider contexts, it cannot be overgeneralised that in the South African context students with disabilities have access to learning based on the availability and provision of AT and devices. A larger study that includes more South African institutions, students with disabilities and Disability Rights Centre staff members, and even academic staff, will be required to confirm the assertion.

Evidence in this study shows that the provision of AT and devices is influenced by limitations among students with disabilities. In literature, however, the International Classification of Functioning (ICF) provides a universal model, where the classification of disability considers both the medical and social aspects of disability, to enable the functionality of those with disabilities in particular, and all diverse persons [72,73]. One would argue that this classification considers both the limitation as imposed by the impairment itself and imposed by the social and physical environment. The Universal Design (UD) model would thus take cognisance of all factors limiting the individual based on the interaction between impairment-related limitations and those caused by the environment. Therefore, the universal model provided by ICF promotes functionality of all diverse persons, by way of creating a totally transformed social and physical environment, that is inclusive to all diversity, including those with disabilities. It is against this background that interventions by way of the UD generally and UDL specifically are proposed for this paper.

## 7. Intervention through Universal Design

The researcher suggests a UD broadly, and UDL specifically, as an intervention that could enhance access to learning for all students through the provision of AT and assistive devices. Context is viewed in light of the understanding that in the contemporary scholarship, both the Global South and the West currently experience common problems, because “…the modern is at crossroads, dominating rather than leading in the domains of knowledges…producing numerous modern problems to which it is not in a position to provide modern solutions” [74] (p. 8). While sensitive to the South African higher education context, particularly that of the specific institution, the researcher’s positionality is informed by the fact that contemporary scholarship concerns itself with finding solutions to common problems through conversation between the Global South and the West. The researcher comes from the perspective that knowledge from different contexts can be brought together and no voice should be stifled because all knowledge has value and is useful [75].

The researcher is suggesting a way of using the UDL, where the provision of AT and devices to aid learning is planned from the outset for all diverse students, and not only for students with disabilities. Other scholars have also proposed that AT should be considered in the UD, if inclusive educational and social environments are to be created for all diverse people [3]. This paper reflects that AT and assistive devices should be incorporated into the UDL, which is more specific to teaching and learning because what is sought to be achieved is that all students with and without disabilities gain access to learning.

The UDL is a teaching approach born from the UD. As the mother framework, the UD is informed by principles, which also guide it. The principles are listed as equitable use, flexibility in use, simplicity and intuition, perceptible information, tolerance of error, low physical effort, and size and space for approach and use [76]. Considering these principles promotes the creation of physical and social environments that cater to all people’s needs, with and without disabilities. The UDL teaching approach also has its core principles, which include flexibility, engagement, representation, action, and expression [69,77]. It emphasises flexible goals, methods, assessments and materials [78]. It involves thinking about and planning for all diverse students, even before they enter the classroom [79] since there is an understanding that human beings have different learning styles and preferences [80]. Such a teaching approach could thus suit all students, including those with disabilities, because it considers all students’ diversity from the start.

The researcher suggests that the provision of AT and assistive devices is planned for all diverse students in advance, not merely for students with disabilities. The provision of AT and devices would then be located within the broader context of teaching, in which the learning needs of all students are considered from the outset. In this instance, what AT and assistive devices would best suit all students’ needs and how they will be provided will be thought about, planned and sourced, before students come to class.

Locating AT and devices within the UDL could thus go a long way in enabling learning because of the principle of flexibility. For example, when computers with JAWS do not read mathematical signs, alternatives could be found easily because flexible goals are emphasised. The core principles speak to consideration of diversity and the active engagement of all diverse students in learning; the UDL optimises learning opportunities and the key to its success is the creation of maximum flexibility from the start [77]. Thus, incorporating the provision of AT and devices into the UDL could go a long way to creating a transformed learning environment that is suitable for all diverse students. This will entail not merely offering individual accommodation for some categories of students, as is typically the case in higher education.

### Possibility for Learning in Using UDL

As highlighted earlier, incorporating the provision of AT and assistive devices into the UDL could enable learning, not only for students with disabilities but for all diverse students at the institution and in South African higher education broadly. While UDL is an American idea by origin, it is not a new teaching strategy in South Africa. Practices and pedagogies aligning with the principles of the UDL are used in special-needs schools in South Africa [81]. It has also been noted that some teachers in inclusive educational settings unknowingly apply the principles of UDL in their teaching [81]. Basic and higher education has porous boundaries, and if UDL is successful in the former, it could also be successful in the latter. It is argued that the UD broadly, and the UDL in particular, are necessary to inform South African higher education curricula if all diverse students are to be included in teaching and learning [8]. Though the South African context in general and the institution of focus are different from the American climate from which the idea of UD and UDL originated, it is influenced by the argument that is all about “pluri-versal, a redemptive and liberatory epistemology that seeks to delink from the tyranny of abstract universals” [82] (p. 13). Therefore, making suitable adjustments where possible could result in an effective intervention despite different contexts.

Intervention through the inclusion of AT and devices in UDL could be seen as a way of soliciting a unified solution to mitigate the challenge of those with disabilities being excluded in both the Global South and West. The provision of AT and assistive devices through UDL could delve deep and further push the boundaries of both worlds, to enhance inclusion broadly and facilitate inclusive education for all, as purported in the UN Convention [2].

## 8. Conclusions

The provision of AT and assistive devices enables students with disabilities to access learning in the programme of Education at the institution. Data avail that the specific institution provides students with disabilities with a number of relevant AT and assistive devices, which these students confirmed promoted their learning. However, the provision is not adequate, and a specific AT and assistive device was inaccessible to one category of disability, consequently limiting learning. It can therefore be concluded that students with disabilities in the Education programme are enabled in accessing learning through the provision of AT and assistive devices. However, some categories are constrained by the kind of AT and devices provided. The researcher thus suggests interventions through the UDL, where the provision of AT and assistive devices is planned and included in the teaching approach for all diverse students from the outset. There is a likelihood of success because the UDL has been tried and proved successful in South Africa. Since the UD and UDL are American ideas that have been successfully implemented in South Africa, it could reflect knowledge being brought together from the South and the West to solve a common global problem, which is a move the contemporary scholarship seeks to achieve.

## Figures and Tables

**Table 1 ijerph-18-03892-t001:** Demographic information of students with disabilities and Disability Rights Centre staff members.

Characteristics		Number
Sex	Male	3
	Female	3
Race	Black	4
	White	2
Age	21–25	2
	26–30	3
	31–40	1
	41 and above	0
Schooling Background	Special Education 3 Mainstream 3	
**Characteristics**		**Number**
Sex	Male	5
	Female	5
Race	Black	6
	White	4
Position in the Disability Unit	Head of Unit	1
	Adaptive technician	1
	Administrators	6
	Sign language Interpreter	1
	Learning disability coordinator	1
Work Experience	0–5 years	3
	6–10 years	3
	11–20 years	3
	21–30 years	1

## Data Availability

The data presented in this study are available on request from the corresponding author. The data are not publicly available due to ethical restrictions.

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
