# Peer review of "Provision of Assistive Technology for Students with Disabilities in South African Higher Education"

_ijerph, 2021, doi:10.3390/ijerph18083892_

Round 1

Reviewer 1 Report

The author takes up important issues, and the article generally meets the criteria of scientific correctness when it comes to individual elements of scientific papers. However, I have doubts as to the consistency of the adopted theoretical concept and research topic. The research itself provides extremely modest material - it is difficult to talk about data saturation. It also seems the author did not present (did not collect) sufficient material to justify his/her conclusions.

Detailed Notes:

Introduction

  1. There is no sound justification for the proposed research topic. Why does the author think that the use of assistive technology can be a source of oppression? I do not find any valid premises justifying the research intentions. This lack significantly affects the course of the research itself and the interpretation of the obtained (modest) material.
  2. “One would further  argue that AT and assistive devices could be one of the ways, in which students with disabilities’ learning in higher education, could be supported and enabled, for them to learn  like all other students without disabilities.” (p.2) – what does it mean, please provide a broader explanation.
  3. “However, though higher education makes provision of AT and assistive devices, students with disabilities continue to be excluded in learning. A number of studies have been conducted to understand the continuous exclusion and this particular study comes from a unique angle, of using decolonial theory, to unveil what is hidden in terms of provision of AT and assistive devices.” (p. 3) – please refer to relevant literature that supports this claim.

Method

  1. “This was peculiar, because literature has consistently revealed that most students with disabilities do not disclose their disabilities on admission and registration in institutions of higher education because they fear stigmatization” (p. 5) – for this statement to be valid, please provide more than just a single resource.
  2. I do not understand the explanations regarding the recruitment of the respondents: 1) if the researcher selected only Education students from the group of 12 people studying Education, Law and Medicine, why did the study group still consist of 12 people; 2) if the researcher focused only on students of Education, why did he/she write that the group comprised of students of other fields of study; 3) author’s justification for selecting students of Education seems to be inadequate to the issues raised in the study.
  3. There is no information about the problems discussed in the interview. Only general issues were presented.
  4. In table 1b, the numbers do not add up to 10 (data missing?). This point should be clarified.
  5. “They were requested to indicate their interest by contacting the researcher through email, and indicating the time convenient for them for interviews, and place they felt comfortable to be interviewed from.” (p. 7) – what places were these?
  6. “ATLAS T could have been used for data analysis but the researcher did not use any software because the transcripts were few.” (p. 7) – A small amount of data is a considerable problem here, as previously mentioned. Why was so limited material collected?

Results

  1. It is not common for research problems to appear only in the results. They were specified on p. 8 and they should appear earlier with a clear reference to relevant theory. At this point - research goals that I believe are: „In the paper, the concept will be used to unveil how coloniality is being perpetuated unknowingly in provision of AT and assistive devices at the institution.” (p. 4); “The concept will help to illuminate the rank of students with disabilities at the institution and provision with AT and assistive devices, as a way of enabling their learning.” (p. 4) and “The coloniality of power is therefore important in this paper to explaining the power dynamics that exist in the provision of AT and assistive devices at the institution.” (p. 5) are specified as research problems: „(a). who are provided with AT and assistive? (b). who decides on the AT and assistive devices to be provided? (c). do AT and assistive devices provided enable students with disabilities’ learning?”, and one can clearly see the inconsistency between the theory and research problems. It is also not common for research problems to appear after the thematic analysis of the material.
  2. I cannot see any validity in the statement, „It emerged from data that it is not all diverse students who are provided with AT and assistive devices but those with disabilities.” (p. 8) – can Disability Rights Centre offer support to other students than those with disabilities?
  3. I suggest not to confuse the author's own results analysis with references to the literature. There is room for the latter in the discussion. I also suggest inspecting the validity of the presented references - are they really relevant for the discussed problems?
  4. The statement: „A student with disabilities who confirmed being provided by a kindle also expressed the same experience that it enabled her to carry books, which she could have not managed to carry because of the category of disabilities she had” (p. 10) is not supported with a quotation/statement made by the respondent.
  5. Perhaps it is worth explaining whether the devices used at universities in Central Africa are new/modern - are there other better ones that do not have the listed disadvantages and limitations? In my view, there are no perfect devices ideally suited to individual needs.

Findings and discussion

I have the impression that the author is trying to “forcefully” adjust his/her research to the theory he/she has selected. Perhaps it is worth considering the validity of the statements: "Coloniality of being exposes the provision of AT and assistive devices to students with disabilities only, as Othering" (p. 11). Do other students need special technologies to a similar extent? Taking into account the specifications of some of them (e.g. Braille displays, Braille notebooks, text readers) - are they really needed by non-disabled students?

Why did students with disability get the "Others" label from the author - do they feel “other”? They were not asked about it.

We do not know how the disability adjudication system works, does it take into account the needs of education, etc. Oppression is created by many factors, among which social attitudes play an important role - perhaps it is worth considering. I also miss another thing here - do students feel stigmatized because of using AT? Does the fact that they are not included in the process of designing AT have any bearing on their negative experiences?

The author refers to the model of social disability (p. 12). I suggest referring to the universal model instead since it is currently the basis for designing activities for people with disabilities. Using universal design is important for people with different needs, nevertheless, in the situation of individual problems caused by deleting/limiting certain functions, it is also important to provide oneself with personal resources that are irreplaceable.

It is worth specifying who works at the Disability Rights Center - are these only non-disabled people?

The author constantly writes about the institution, but the learning process, especially at the university level, is mainly the student's own activity undertaken not only in institutional conditions. In addition, the institutional environment comprises many factors, such as legal and organizational policy, attitudes of specific groups of people, etc., and all of this may be important for the psychosocial and educational functioning of students with disabilities.

Author Response

Dear Reviewer

Please find my response to your comments for my manuscript

Kind regards

Sibonokuhle Nlovu

Reviewer 2 Report

Thank you for inviting me to review the very interesting paper: "Provision of assistive technology for students with disabilities in South African higher education: A decolonial perspective". The paper brings a new perspective in the understanding of support provision for students with disabilities, highly relevant for the understanding of the cultural status quo.

Nevertheless there are methodological issues:

There is a need to make the research question explicit. The method chosen should be a consequence of your research question.

Clarify recruitment process and number of students involved. Process is not clear. 12 student at the beginning, how many from education? Does a sampling from the same university and same department give generalizable data to understand South African context?

There is no information about the methodology chosen for the conversation with students (semi-structured interview is good input, but what are the structured themes used in conversation?).

Table 2 needs to be checked in relation to total number of participants.

How was data collected? How were transcripts and recording done? Which were the topics used during qualitative interviews? How consensus was reached within the 3 different readers?

How much data was retrieved if there was no need to use a software to analyze it?

Direct quotations regarding respondents conversations would increase readability.

I would have a provocatory question why Universal Design for learning could not be considered a form of cultural colonialism (indirect in case of South Africa) from the monocultural North (Carolinian) American perspective?

Author Response

Dear Reviewer

Please recieve my response to your comments on my manuscript.

I hope you find everything in order.

Kind regards

Sibonokuhle Ndlovu

Round 2

Reviewer 1 Report

Detailed Notes:

Introduction

  1. There is no sound justification for the proposed research topic. Why does the author think that the use of assistive technology can be a source of oppression? I do not find any valid premises justifying the research intentions. This lack significantly affects the course of the research itself and the interpretation of the obtained (modest) material.

The paper is written at the time when the issue of decolonisation in higher education in South Africa broadly is topical, and as such seeking to decolonise pedagogy is one of the issues scholars find it important to engage debates on, research on and write about. In this particular paper, the author is concerned about identifying what is colonial in teaching and learning, with specific reference to provision of AT, as one way of accessing learning by students with disabilities.

Using the decolonial theory lens, provision that can isolate other social groups and make them a ‘special’ group, is considered colonial.  While provision of assistive devices can aid learning by students with disabilities, in a situation where there is a specific structure that provides to those students alone and not any other student, AT and assistive devices are seen as perpetuating isolation, they discriminate, they stigmatise and there is labelling of the particular students, by those making a provision. It is from this background that I interpret such a kind of provision as oppressive. The oppression is invisible and may not be seen at surface level, but only with decolonial spectacles.

I thus justify my proposed research topic  and the theoretical lens and perspective from which I come from, which makes me justify  that in the particular instance and context, use of assistive technology can be a source of oppression. I saw decolonial theory as unveiling of the hidden, that which may not be seen at surface level. Oppression through provision of assistive devices to students with disabilities may not be seen at surface level but only through unveiling of the hidden through decolonial theory.

I understand that the author tried to refer to the decolonization perspective in her work, she explained it theoretically. However, I do not see any evidence, in the first and the revised versions, that this perspective is justified in her research. The author has not proved in the collected material that students with disabilities experienced oppression in its various forms. I still notice the problem of the inconsistency of theoretical assumptions or even final conclusions with research problems and research results. Unfortunately, I find the views presented by the author in her work and in responses to the comments unconvincing, taking into account references to literature, the research concept, and its results.

  1. “One would further argue that AT and assistive devices could be one of the ways, in which students with disabilities’ learning in higher education, could be supported and enabled, for them to learn like all other students without disabilities.” (p.2) – what does it mean, please provide a broader explanation.

What the author means is that indeed provision of AT and assistive devices could be one of the ways, in which students with disabilities’ learning in higher education, are  supported and enabled, for them to learn like all other students without disabilities. Despite the limitations imposed by impairments on students with disabilities, when provided by AT and assistive devices, they could access learning as does those students without disabilities.

The author has backed this with literature which include Mji, MacLachlan, Melling-Williams & Gcaza  (2009) Matter, Harniss,  Oderud, Borg J, Eide, (2017), Lyner-Cleophas, M. (2019). Kisanga, & Kisanga (2020).  

Highlighting the phrase in the quote (omitted by the author): "like other students without disabilities" was to draw the author's attention to this issue, which is important here and appears throughout the work: what does it mean that students with disabilities learn like non-disabled students? In what sense? What specific solutions can create a university environment that would meet the principles of universal design?

3.“However, though higher education makes provision of AT and assistive devices, students with disabilities continue to be excluded in learning. A number of studies have been conducted to understand the continuous exclusion in learning in South African higher education. This particular study comes from a unique angle of using decolonial theory, to unveil what is hidden in terms of provision of AT and assistive devices.” (p. 3) – please refer to relevant literature that supports this claim.

The claim that students with disabilities continue to be excluded in teaching and learning in the South African higher education has been validated through literature (Crous, 2004; Sukhray-Ely, 2008; Kajee, 2010; Mutanga, 2017).

Method

  1. “This was peculiar, because literature has consistently revealed that most students with disabilities do not disclose their disabilities on admission and registration in institutions of higher education because they fear stigmatization” (p. 5) – for this statement to be valid, please provide more than just a single resource.

Three sources that validate the statement on students with disabilities not disclosing their disabilities on admission and registration has been added. They include   Kendal (2016), Zaussinger, S & Terzieva, (2018) and  Eccles,  Hutching, Hunt & Heaslip, (2018).

Unfortunately, I do not see these references in the revised article.

2.I do not understand the explanations regarding the recruitment of the respondents: 1) if the researcher selected only Education students from the group of 12 people studying Education, Law and Medicine, why did the study group still consist of 12 people; 2) if the researcher focused only on students of Education, why did he/she write that the group comprised of students of other fields of study; 3) author’s justification for selecting students of Education seems to be inadequate to the issues raised in the study.

The author has clearly explained the recruitment of respondents by only making reference to students who studied Education and Disability Rights Centre staff members, who were participants sampled for this particular paper (p.5- 6, 225-238).

The author has not made mention of other students Medicine and Law who did not form part of this particular paper.

3.There is no information about the problems discussed in the interview. Only general             issues were presented.

The problem of the study has been included that though South African higher education makes effort to provide AT and assistive device to students with disabilities, to enable them to learn, the way provision is done, defeats the whole purpose for it is intended. By virtue, students with disabilities continue to confront limitations in their learning from some AT and assistive devices provided (p. 3, line 122-125)

4.In table 1b, the numbers do not add up to 10 (data missing?). This point should be clarified.

The issue of numbers on table 1b has been corrected and clarification has been made as to why the numbers add to those provided. The author has clearly put it that the sample included 16 participants (N=16), of which six were students with disabilities in the programme of Education and 10 were Disability Rights Centre staff members, who  all volunteered to participate (p.5- 6, line 225-238).

5.“They were requested to indicate their interest by contacting the researcher through email, and indicating the time convenient for them for interviews, and place they felt comfortable to be interviewed from.” (p. 7) – what places were these?

The student participants were interviewed from different places they felt were convenient for them. One student with hearing impairments was interviewed from the library loans, which was a very quiet space. Three of them were interviewed from their residences because they felt it was convenient for them as they had physical and visual limitations, respectively. One chose to be interviewed at a place outside the dining hall on the reason that she liked the place, and one was interviewed in his office, as he was also a staff member. Disability Rights Centres staff member were interviewed from their offices (p. 298-305).

  • “ATLAS T could have been used for data analysis but the researcher did not use any software because the transcripts were few.” (p. 7)

There were only 16 interview transcripts. The author understood that it was her responsibility to do the analytic work, and it was manual analysis that would enable her a deeper understanding of data as she read it repeatedly. Besides, manual analysis would allow for coding of data and deciding which codes link with the broader objectives, aims and the research questions. This would allow for meaningful (and not automated) labelling and categorisation of data. Repeated manual scouring of data was necessary in order to interrogate it thoroughly, and to understand it broadly.

The remark was not about the analytical method but about its quantity – this was stressed in the first review.

 A small amount of data is a considerable problem here, as previously mentioned. Why was so limited material collected?

Students with disabilities are few in terms of student population in South African institutions of higher education. They usually make 0,5 of student population (Fotim Report, 2011). However though the sample was small, it was representative. Representativity is the extent to which the substantive interest of the study is adequately represented in the sample (Babbie, 1973). The sample had participants who are key, and had sources of rich data. Students with disabilities and Disability Right Centre staff members provided rich data on the phenomena of interest to the study. The sample cut across varieties of race, age, gender, disability categories, work experiences, positions of authority, as well as two levels of study, namely under-graduate and post-graduate. The specific variables balanced the representation

It is not about the amount of collected data, the number of students, or even their representativeness. The research is, I believe, qualitative. It is important to collect material that will allow the author to exhaust the assumed problems; to fully prove the theory that the author has proposed. Representativeness cannot be assessed in the sense that the author understands it (Representativity is the extent to which the substantive interest of the study is adequately represented in the sample), because there is little reference to the obtained data that would illustrate the research problem.

I do not understand why the author writes, “Disability categories of students and of those members, who were disabled, have been purposely left out. The researcher views stating the disabilities as perpetuating labelling and segregation that she is so much opposed to”, if on pp. 1, 2 she states, „One would argue that depending on the category of disabilities and their severities, AT and assistive devices can enable and empower peoples’ functionality generally,  and more particularly persons with disabilities, who usually confront challenges of  functionality as resulting from their impairments” and “Provision of AT and assistive devices could be one of the ways, in which students with disabilities’  learning in higher education.” The category of disability is therefore important. Disability is a fact that is important on an individual and social level and as such requires both individual and environmental solutions. I have a feeling that the author omits this first important point. I also find no reference to the universal model represented in the ICF, although the author states that she did so in response to the review. “Again the focus of the study is on provision of AT and assistive devices, which has nothing associated with the disabilities of the participants in particular” – what are they associated with then? This is inconsistent with the author's previous statements pp. 1 and 2 (abovementioned).

Results

  1. It is not common for research problems to appear only in the results. They were specified on p. 8 and they should appear earlier with a clear reference to relevant theory. At this point - research goals that I believe are: „In the paper, the concept will be used to unveil how coloniality is being perpetuated unknowingly in provision of AT and assistive devices at the institution.” (p. 4); “The concept will help to illuminate the rank of students with disabilities at the institution and provision with AT and assistive devices, as a way of enabling their learning.” (p. 4) and “The coloniality of power is therefore important in this paper to explaining the power dynamics that exist in the provision of AT and assistive devices at the institution.” (p. 5) are specified as research problems: „(a). who are provided with AT and assistive? (b). who decides on the AT and assistive devices to be provided? (c). do AT and assistive devices provided enable students with disabilities’ learning?”, and one can clearly see the inconsistency between the theory and research problems. It is also not common for research problems to appear after the thematic analysis of the material.

The sub research questions were removed from the section on results and placed in the section under introduction, together with the main research question of the study (p. 3, 126-128).

  1. I cannot see any validity in the statement, It emerged from data that it is not all diverse students who are provided with AT and assistive devices but those with disabilities.” (p. 8) – can Disability Rights Centre offer support to other students than those with disabilities?

Provision of AT and assistive devices to students with disabilities was by The Disability Rights Centre. This is a structure specifically for supporting students with disabilities. It might not have been expected to have also provided AT and assistive devices to students without disabilities who might have not needed them.

The whole argument  from the coloniality of being  perspective is that the presence of the structure that provides for a particular category of students (those with disabilities) reflects that there is lack of a complete  transformed institutional environment in which all diverse students’ learning needs, including those with disabilities are met in the mainstream. If there was a full transformed inclusive environment there wouldn’t be any need of a structure as Disability Right Centre that make provision of AT and assistive devices only to those with disabilities.

The problem is that the author does not prove this point in her research.

  1. I suggest not to confuse the author's own results analysis with references to the literature. There is room for the latter in the discussion. I also suggest inspecting the validity of the presented references - are they relevant for the discussed problems?

The reviewer’s comment has been considered and the author’s own results analysis and reference to literature has been separated. The presented literature that confirms the results were included in the discussion. The author was also careful to consider literature that is relevant to the discussion.

The author still left many references to literature in the discussion.

4.The statement: A student with disabilities who confirmed being provided by a kindle       also expressed the same experience that it enabled her to carry books, which she could have not managed to carry because of the category of disabilities she had” (p. 10) is not supported with a quotation/statement made by the respondent.

The statement made by the student has been included as suggested by the reviewer. The words were inserted as follows: ‘They helped me by giving me a kindle which allows me to download books and I don’t have to carry heavy books around’

Unfortunately, I do not see this quotation in the revised manuscript.

  1. Perhaps it is worth explaining whether the devices used at universities in Central Africa are new/modern - are there other better ones that do not have the listed disadvantages and limitations? In my view, there are no perfect devices ideally suited to individual needs.

As the author is focused on South Africa, which is not in Central Africa, the response will focus on Southern Africa and South Africa specifically.

In Southern Africa largely and South Africa specifically, provision of AT and assistive devices mainly focusses on mobility, hearing, vision communication and cognition limitations. Examples of AT and assistive devices provided thus include among others, include prosthetics, hearing aids, spectacles, white canes and wheelchairs  One would argue that various types of  both low  and high-tech  AT and assistive devices are provided in institutions of higher education in the region broadly and in  South Africa specifically.  The paper however discusses provision of AT and assistive devices in the way of enabling access to learning, in which some of the listed do play a role.

Findings and discussion

I have the impression that the author is trying to “forcefully” adjust his/her research to the theory he/she has selected. Perhaps it is worth considering the validity of the statements: "Coloniality of being exposes the provision of AT and assistive devices to students with disabilities only, as Othering" (p. 11). Do other students need special technologies to a similar extent? Taking into account the specifications of some of them (e.g. Braille displays, Braille notebooks, text readers) - are they really needed by non-disabled students?

Students without disabilities would not need Assistive Technology as those with disabilities. What the author seeks to bring out  from the lens of coloniality of being is that the presence of  the Disability Rights Centre, providing AT only to a particular group of students reflects an environment that is exclusive; where other students, more specifically those with disabilities are ‘Othered’. An inclusive institutional environment would not even need to have such a structure.   All diverse needs of all students, including those with disabilities will be met in a fully transformed learning environment. The author has made effort to clarify this in the paper. 

Neither the posed research questions nor the obtained results confirm such conclusions. The following statement,  “An inclusive institutional environment would not even need to have such a structure.   All diverse needs of all students, including those with disabilities will be met in a fully transformed learning environment” has not been verified or confirmed in the author's research. The research concerned a totally different topic. 

The reviewer understands that non-disabled students do not need any special resources in the learning process. Why then does the author formulate statements that such measures are applied/granted only to people with disabilities - is it not obvious?

Why did students with disability get the "Others" label from the author - do they feel “other”? They were not asked about it.

From the coloniality of being perspective, students with disabilities are viewed as the Other from the categorisation process in which ‘normalcy’ is used as the standard by the dominant society. By way of difference in body they fall outside the margin of ‘normal’. It is from this background that the author uses the term Other in the paper because she uses decolonial theory as the torchlight. Otherwise the students themselves were not asked about it, whether they feel being the Other or not.

We do not know how the disability adjudication system works, does it take into account the needs of education, etc. Oppression is created by many factors, among which social attitudes play an important role - perhaps it is worth considering. I also miss another thing here - do students feel stigmatized because of using AT?

Students themselves in the study do not feel stigmatised. Stigmatisation only comes in when the provision  of AT and assistive devicesis looked at from the deeper perspective of coloniality of being, when there is a specific structure as the Disability Right Centre, meant to provide students with disability as a separate group. That’s what the author is arguing in the whole paper.

Students’ experiences should be the basis for this deeper perspective. Inference about stigmatization based on the existence of a certain structure (Center) is an overinterpretation here.

Does the fact that they are not included in the process of designing AT have any bearing on their negative experiences?

Yes, exclusion in designing the AT has a bearing on their negative experiences. Had they been involved, they would not have experiences of Assistive Technology, which were not working for them like the computers with JAWS that did not read mathematical signs. Had they been involved, that limitation would be detected early and rectified before the gadgets were dispatched.  When they are not involved, people without a lived experience of disability design Assistive Technology that do not suit the individual needs for whom that AT is designed for. Excluding those with disabilities in what involves them is highly contested in the disability scholarship and those with disabilities themselves state that its limiting them who have a lived experience, hence the slogan, ‘Nothing for us, without us’   

The author refers to the model of social disability (p. 12). I suggest referring to the universal model instead since it is currently the basis for designing activities for people with disabilities. Using universal design is important for people with different needs, nevertheless, in the situation of individual problems caused by deleting/limiting certain functions, it is also important to provide oneself with personal resources that are irreplaceable.

The suggestion of the reviewer was taken and the author referred to Universal model instead of the social model

I do not see this reference in the revised manuscript.

It is worth specifying who works at the Disability Rights Center - are these only non-disabled people?

The author has specified that there are both non-disabled and disabled staff members at the Disability Rights Centre and the numbers provided.

Why then does the author state that people with disabilities are not included in the process of designing AT  (i.a. “Of interest is  that those who determine the kind of AT and assistive devices students with disabilities require, are they themselves not disabled)? Is there evidence for this claim in the author's research?

The author constantly writes about the institution, but the learning process, especially at the university level, is mainly the student's own activity undertaken not only in institutional conditions. In addition, the institutional environment comprises many factors, such as legal and organizational policy, attitudes of specific groups of people, etc., and all of this may be important for the psychosocial and educational functioning of students with disabilities.

I agree with the reviewer that learning at university level is mainly the student activity and responsibility.  However, the institution also plays a great role in the learning of students, more specifically those with disabilities, who have impairment needs that need to be met by the institution. Thus, the responsibility of learning cannot be solely left to students.  For students   to access learning, an institution should be seen to be playing its role, by transforming the learning environment to enable all students’ learning, including those with disabilities. It is for this reason that the author has continually referred to the institution.  

The responsibility for learning does not rest with the students alone. Learning is a multifaceted process taking place in a complex material and personal environment, as previously mentioned in the comment. There was no reflection in the research project on this important fact, although the author writes, referring to literature, about attitudes in the academic environment.

I do not deny the value of the theories, which, as it seems, are dear to the author and, as she writes, important in educational practice. Nevertheless, I maintain my position that the author tries to find evidence in the collected material to confirm a certain theory she feels is true. At the same time, she ignores the obtained empirical evidence, such as the positive aspects of using AT for learning reported by students. I do not see here the consistency of the theory with the research and its results, but I see numerous overinterpretations, simplifications, and generalizations that are not grounded in the collected material.

I think that the author is aware of the fact that the undertaken project does not reflect the essence of the matter when she writes, “While the institution is making significant efforts of providing even  the most expensive AT and an assistive device, as the eye tracker, it is not addressing the deep underlying cause of exclusion of those with disability, which is coloniality. The cause of exclusion is deeper than seen at surface level. The structures, practices and processes exclude those who are considered as not ‘normal’ and in this case, even when AT and assistive are provided, they are still exclusive one way or the other, hence some of them, do not enable learning but constrain it.”

I encourage the author to prepare a study in which she would specify the assumptions of universal design and the possibilities of its implementation in the education system in her country.

Author Response

Dear Reviewer

Please find attached my responses for your second round of comments. In response to your major comments in which you felt that i was forcing my theory on data, i had to change the whole theory and used a more relevant one. I strongly feel that this time data and evidence speak for themselves and there is consistency throughout the papers. In my responses, i have italicised all that i have removed and my new responses are in red.

Thank you very much for your very constructive comments, which i have learnt a lot about doing a consistent research, that is based on results and evidence.

Kind regards

Sibo

Reviewer 2 Report

Thank you for adapting the manuscript to the comments of the reviewers and for the extensive explantions.

I would only have a couple of remarks:

Direct quotations "Citation..." regarding respondents conversations would increase readability.

I have a provocatory question that would make the case stringer for Universal Design.  Why Universal Design for learning should not be considered a form of cultural colonialism (indirect in case of South Africa) from the monocultural North (Carolinian) American perspective?

Author Response

Dear Reviewer

Please find attached my responses to your second round of review comments. Your comments were so valuable to my paper and helped me greatly in my research work generally. It has been great working with you.

Kind regards

Sibo
